# Lipase-Catalyzed Phospha-Michael Addition Reactions under Mild Conditions

**DOI:** 10.3390/molecules27227798

**Published:** 2022-11-12

**Authors:** Yuelin Xu, Fengxi Li, Jinglin Ma, Jiapeng Li, Hanqing Xie, Chunyu Wang, Peng Chen, Lei Wang

**Affiliations:** 1Key Laboratory of Molecular Enzymology and Engineering of Ministry of Education, School of Life Sciences, Jilin University, Changchun 130023, China; 2State Key Laboratory of Supramolecular Structure and Materials, Jilin University, Changchun, 130023, China; 3The Second Hospital of Jilin University Changchun, Jilin University, Changchun 130041, China

**Keywords:** lipase, biocatalysts, promiscuity, C-P bonds, Michael addition reactions

## Abstract

Organophosphorus compounds are the core structure of many active natural products. The synthesis of these compounds is generally achieved by metal catalysis requiring specifically functionalized substrates or harsh conditions. Herein, we disclose the phospha-Michael addition reaction of biphenyphosphine oxide with various substituted β-nitrostyrenes or benzylidene malononitriles. This biocatalytic strategy provides a direct route for the synthesis of C-P bonds with good functional group compatibility and simple and practical operation. Under the optimal conditions (styrene (0.5 mmol), biphenyphosphine oxide (0.5 mmol), Novozym 435 (300 U), and EtOH (1 mL)), lipase leads to the formation of organophosphorus compounds in yields up to 94% at room temperature. Furthermore, we confirm the role of the catalytic triad of lipase in this phospha-Michael addition reaction. This new biocatalytic system will have broad applications in organic synthesis.

## 1. Introduction

The incorporation of phosphorus can beneficially modify the physicochemical and biological properties of bioactive molecules. Organophosphorus compounds are increasingly utilized in medicine and biological systems because phosphorus substituents can improve the pharmacokinetic and pharmacological properties of drugs, including their potency, cell permeability, and metabolic stability [1,2,3,4,5]. The phospha-Michael addition of phosphorus nucleophiles is one of the most practical approaches to generating new C-P bonds and transforming them into corresponding 1,4-addition products. [6,7,8,9,10,11,12].

Over the past decades, a general method for the conjugate transfer of phosphinyl groups from silylphosphines to α-β-unsaturated carbonyl catalyzed by a bench-stable palladium (II) catalyst has been reported (Figure 1a) [13,14]. Song and co-workers reported the protocol for hydrophosphination of α-β-unsaturated compounds with diphe-nylphosphine catalyzed by nickel or palladium (Figure 1b) [15,16,17,18,19]. Organic catalysts can mediate the formation of the phosphorus–carbon bond between biphenylphosphine oxide and the Michael acceptor, as reported by Bharati’s group (Figure 1c); these guanidine derivative catalysts exhibit better catalytic ability than triethylamine [20,21,22]. Despite this progress, inconvenient and non-atom-economic substrates (such as silylphosphines), expensive metal catalysts, and hazardous organic solvents limit the practical application of the methods. Therefore, the development of new phospha-Michael addition reactions under mild conditions is still an urgent issue.

Enzymes, which catalyze metabolism in organisms, have the advantages of high catalytic efficiency and excellent selectivity and are more environmentally attractive than traditional chemical catalysts. In general, the most impressive enzymes are thought to only catalyze specific substrates from a natural selection; this property is called specificity, which limits the application of enzymes in industrial production. By contrast, “catalytic promiscuity” occurs when the catalytic abilities of the active site are used to catalyze a distinctly different type of reaction. Currently, hydrolase, hemoproteins, and artificial metalloenzymes have emerged as promising catalysts for non-natural reactions forming C-C and C-hetero bonds [23,24,25,26,27,28,29,30]. Lipases (EC 3.1.1.3) are the ideal biocatalyst due to their commercial availability, broad substrate specificity, and good stability in nonaqueous media. As one of the interfacial enzymes, the interfacial activation of lipases results primarily from conformational changes in the enzymes that expose the active site and provide a hydrophobic surface for interaction with the substrate [31]. These properties make lipases even more versatile in the rapidly developing field of catalytic promiscuity [32,33]. Our group and other researchers have previously reported that lipases can promote numerous promiscuous reactions, such as the Michael addition, aldol reaction, and oxidation reactions [34,35,36,37,38,39,40,41]. Compared with previous chemical methods, lipase-catalyzed non-natural reactions have many advantages, such as mild reaction conditions, environmental friendliness, and a green, reusable biocatalyst. As a continuation of our recent research exploring new non-natural reactions of lipase in organic synthesis, we have studied the reaction of dimethylphosphite with β-nitrostyrenes or benzylidenemalononitrile. In this work, we establish a convenient and efficient synthesis of organophosphorus compounds via a lipase-catalyzed phospha-Michael addition (Figure 1d).

## 2. Results and Discussion

Initially, we selected β-nitrostyrene (**1a**) and biphenyphosphine oxide (**2**) as the model substrates for the lipase-catalyzed phospha-Michael addition reaction to optimize the conditions (Table 1). All of the selected lipases can catalyze the phospha-Michael addition reaction (entries 1–4). Novozym 435 (a commercial immobilized Cal-B) was the best at generating a yield of the product in 2 h (**3a**, entry 1), and the other lipases demonstrated relatively satisfactory yields in 4 h (entries 2–4). The above results are in accordance with the previous works about lipase-catalyzed Michael additions [42,43]. The catalytic ability of these lipases was higher than that of chemical catalysts (triethylamine, entry 5). Phenylmethanesulfonyl fluoride (PMSF) is usually used as an irreversible inhibitor of lipase for mechanism studies [44]. In this phospha-Michael addition reaction, Novozym 435 was inactivated by PMSF to confirm the role of the active center of the lipase, and the reaction did not proceed when the inactive lipase was used as the catalyst (entry 7). Similar results were observed in the thermally denatured lipase-catalyzed reaction and the control reaction (entries 8, 9). This result coincides with the lipase-catalyzed aza-Michael addition previously reported by us [45].

An appropriate reaction medium can notably improve the yields of enzymatic reactions [46]. Thus, various solvents were evaluated for the lipase-catalyzed Michael addition (Table 1, entries 1, 9–16). Novozym 435 showed good catalytic activity in polar solvents, such as MeCN, EtOH, and MeOH, but poor performance in dichloromethane, toluene, and hexane. However, the reaction did not show a satisfactory yield due to the poor solubility of the substrate in water (entry 14). Based on these results, polar solvents may help maintain the catalytic performance of enzymes. Novozym 435 showed the highest yield in EtOH (entry 1). Thus, EtOH was selected as the solvent in this reaction. Meanwhile, using green and low-cost media makes this method more environmentally friendly and economical than other synthesis processes for this reaction [47]. We then monitored the enantioselectivity for this phospha-Michael addition reaction. Product **3a** was analyzed by chiral HPLC analysis. Unfortunately, neither triethylamine nor lipase can provide chiral induction in the reaction.

We also evaluated the effect of the enzyme dosage (Table 1, entries 1, 17, and 18). When the catalyst dosage varied from 150 U to 300 U, the yield increased with the increasing catalyst dosage. However, a higher enzyme dosage (>300 U) did not further increase the reaction yield. Therefore, from an economical point of view, 300 U was chosen as the optimum enzyme dosage. Meanwhile, the time course of the lipase-catalyzed phospha-Michael addition reaction was investigated in the Appendix A.

Enzyme immobilization has been found to be a solution for many enzyme limitations: it can improve enzyme stability; alter enzyme selectivity, specificity, and activity; reduce inhibitions; enlarge the operation window and resistance to chemicals; and even be coupled with enzyme purification [48,49,50,51,52,53,54,55,56,57]. In this work, the reusability of Novozym 435 was investigated; the Novozym 435 was able to be recovered by filtration easily, washed with dichloromethane, and then reused directly. Figure 1 shows that as the number of cycles increases, the yield of **3a** decreases; it obtained a 79% yield even after seven cycles. The decrease in the yield of **3a** might be due to the leakage of proteins from the support and the slight dissolution of the support during the reaction cycles (Appendix A).

Under the optimum reaction conditions for lipase-catalyzed phospha-Michael addition reactions (Table 1, entry1), various β-nitrostyrene derivatives (**1**) and biphenyphosphine oxide (**2**) were utilized to evaluate the scope of this enzymatic method (Table 2). When electron-donating substituents, such as methyl and methoxy groups, were attached to the phenyl ring, products **3b–3d** and **3j** were obtained in high yields. When the substituents on the phenyl ring were electron-withdrawing, such as Cl– and Br–, products **3e–3i** were also obtained in satisfactory yields. Operation of this method is simple and practical, enabling the precipitation of the product isolated by filtration and washed with cold acetonitrile to obtain **3**. The method also has excellent yields and diverse libraries of target compounds with potential biological activities.

Inspired by the above experiment, further examination of the substrate scope of the lipase-catalyzed phospha-Michael addition reaction was performed. Biphenyphosphine oxide (**2**) and benzylidene malononitrile (**4**) were chosen to evaluate the scope of this protocol (Table 3). The products have high yields regardless of whether the electron-withdrawing substituents or electron-donating substituents are attached to the phenyl of **4** (**5a–5i**). For the other aromatic rings, products **5j** and **5k** were also generated in good yields. These results demonstrate the broad substrate applicability of the lipase-catalyzed phospha-Michael reaction.

A possible reaction pathway for the lipase-catalyzed phospha-Michael addition reaction was speculated based on our preliminary results and previous reports (Figure 2). First, 2a is deprotonated by the catalytic triad of lipase. Subsequently, a nucleophilic attack on nitroalkene can generate a zwitterionic intermediate I with high reactivity, which might be stabilized by the oxyanion of lipase. Finally, the protonation of intermediate I in the presence of the Asp-His dyad can provide the desired product 3a accompanied by the regeneration of the reactive lipase active site.

## 3. Materials and Methods

### 3.1. General Information

PPL (porcine pancreas lipase, 5600 U/g), PSL (*Pseudomonas sp.* lipase, 8500 U/g), CalB (Lipase B from *Candida antarctica*, 10,000 U/mL), and CSL (*Candida sp.* lipase, 6400 U/g) were purchased from Shanghai Yuan Ye Biological Technology Company and Novozym 435 (15,000 U/g) was purchased from Sigma-Aldrich China Co. (Beijing, China). One unit of the enzyme activity was defined as the amount of enzyme required to hydrolyze 1 μmol of p-nitrophenyl acetate per minute at 30 °C. All other chemical reagents were purchased from commercial suppliers (Bide Pharmatech, Aladdin, Energy Chemical). All of the commercially available reagents and solvents were used without further purification. Proton nuclear magnetic resonance (1H NMR) spectra were recorded on a 400 MHz spectrometer in CDCl_3_ or DMSO. Chemical shifts of protons are reported in parts per million downfield from tetramethyl silane (TMS) and are referenced to residual protium in the NMR solvent (CHCl_3_ = δ 7.26 ppm, DMSO = δ 2.50 ppm). NMR data are presented as follows: chemical shift (δ ppm), multiplicity (s = singlet, d = doublet, t = triplet, q = quartet, m = multiplet, br = broad), coupling constant in Hertz (Hz), integration. The experiments were performed in triplicate, and all data were obtained based on the average values.

### 3.2. General Procedure for the Synthesis of ***3***

To create a mixture of β-nitrostyrene (**1**, 0.5 mmol) and biphenyphosphine oxide (**2**, 0.5 mmol) in EtOH (1 mL), lipase (protein content: 300 U) was added and stirred in a room temperature shaker until completion of the reaction as indicated by TLC. The resulting precipitate was then filtered and washed with cold ethanol to obtain the compounds of **3** as pure products requiring no further purification. The products were further characterized by NMR spectroscopy.

### 3.3. General Procedure for the Synthesis of ***5***

To create a mixture of benzylidene malononitrile (**4**, 0.5 mmol) and biphenyphosphine oxide (**2**, 0.5 mmol) in EtOH (1 mL), lipase (protein content: 300 U) was added and stirred in a room temperature shaker until completion of the reaction as indicated by TLC. After completion of the reaction, the reaction mass was extracted with ethyl acetate (3 × 5 mL). The organic layers were combined and subjected to drying by a rota-evaporator to obtain crude **5**, which was purified using column chromatography (hexane-EtOAc).

### 3.4. Synthetic Procedures of Benzylidene Malono-Nitrile ***4***

Benzaldehyde (10 mmol), malononitrile (9 mmol), and DABCO (0.1 mmol) were added to a round-bottom flask with a 25 mL capacity, followed by the addition of 10 mL of water, and stirred in a room temperature shaker for 12 h. Upon completion of the reaction, as indicated by TLC, a precipitate came out from the mixture. The precipitate was collected by filtration, washed with n-hexane several times, and dried to obtain benzylidene malononitrile (**4**).

### 3.5. The Denaturation of Novozym 435 Treated by PMSF

Novozym 435 (20 mg/mL) was incubated in ethanol with the PMSF (phenylmethanesulfonyl fluoride) stock solution (10 mmol/L in ethanol) at room temperature to denature the lipase. The reaction system was filtered, and the residue was washed with ethanol to remove the unbound PMSF. Then, the denatured Novozym 435 was dried overnight under a vacuum.

#### 3.5.1. **3a** ((2-nitro-1-phenylethyl) diphenylphosphine oxide)

^1^H NMR (400 MHz, DMSO-d6) δ 8.18–8.10 (m, 2H), 7.78–7.70 (m, 2H), 7.68–7.62 (m, 3H), 7.41 (td, J = 7.2, 1.6 Hz, 3H), 7.37–7.31 (m, 2H), 7.23–7.12 (m, 3H), 5.15–4.97 (m, 2H), 4.91–4.81 (m, 1H). ^13^C NMR (151 MHz, DMSO) δ 133.20, 132.62, 131.52, 131.21, 131.15, 131.06, 131.00, 130.62, 130.41, 129.81, 129.73, 128.98, 128.90, 128.83, 128.37, 76.47, 76.06, 44.44, 44.02. ^31^P NMR (243 MHz, DMSO) δ 30.84.

#### 3.5.2. **3b** ((2-nitro-1-(p-tolyl) ethyl) diphenylphosphine oxide)

^1^H NMR (400 MHz, Chloroform-d) δ 8.01–7.94 (m, 2H), 7.67–7.56 (m, 3H), 7.51–7.45 (m, 2H), 7.40 (dd, J = 7.2, 1.6 Hz, 1H), 7.30 (td, J = 6.4, 5.6, 3.6 Hz, 2H), 7.17 (dd, J = 8.0, 2.0 Hz, 2H), 7.02 (d, J = 7.6 Hz, 2H), 5.11–5.01 (m, 1H), 4.77–4.71 (m, 1H), 4.41 (s, 1H), 2.26 (d, J = 1.6 Hz, 3H). ^13^C NMR (151 MHz, CDCl_3_) δ 135.71, 130.35, 129.74, 128.85, 128.79, 128.69, 128.23, 128.09, 127.59, 127.40, 127.15, 126.91, 126.05, 125.97, 114.09, 74.98, 74.77, 74.56, 73.57, 73.53, 43.28, 42.86, 18.73. ^31^P NMR (243 MHz, CDCl_3_) δ 24.87.

#### 3.5.3. **3c** ((1-(4-methoxyphenyl)-2-nitroethyl) diphenylphosphine oxide)

^1^H NMR (400 MHz, DMSO-d6) δ 8.15–8.07 (m, 2H), 7.81–7.72 (m, 2H), 7.65 (dt, J = 6.0, 3.6 Hz, 3H), 7.45–7.32 (m, 5H), 6.80–6.74 (m, 2H), 5.06–4.90 (m, 2H), 4.84–4.77 (m, 1H), 3.67 (s, 3H). ^13^C NMR (151 MHz, DMSO) δ 159.06, 133.08, 132.53, 131.18, 131.07, 130.28, 129.75, 129.67, 128.99, 128.91, 124.34, 114.20, 76.34, 55.38, 55.06, 43.58, 43.15. ^31^P NMR (243 MHz, DMSO) δ 30.56.

#### 3.5.4. **3d** ((1-(4-isopropylphenyl)-2-nitroethyl) diphenylphosphine oxide)

^1^H NMR (400 MHz, Chloroform-d) δ 8.01 (t, J = 9.2 Hz, 2H), 7.66 (dq, J = 14.0, 7.2 Hz, 3H), 7.43 (q, J = 7.6, 6.4 Hz, 3H), 7.31 (d, J = 2.8 Hz, 1H), 7.28 (s, 1H), 7.21 (d, J = 8.0 Hz, 2H), 7.09 (d, J = 7.2 Hz, 2H), 5.13 (s, 1H), 4.78 (d, J = 13.6 Hz, 1H), 4.45 (d, J = 10.8 Hz, 1H), 2.89–2.79 (m,1H), 1.20 (d, J = 6.8 Hz, 6H). ^13^C NMR (151 MHz, CDCl_3_) δ 149.03, 132.73, 132.53, 132.04, 131.31, 131.14, 130.76, 130.47, 129.81, 129.35, 129.27, 128.61, 128.28, 128.21, 126.84, 75.89, 75.71, 45.77, 45.33, 33.68, 23.82. ^31^P NMR (243 MHz, CDCl_3_) δ 30.26.

#### 3.5.5. **3e** ((1-(4-fluorophenyl)-2-nitroethyl) diphenylphosphine oxide)

^1^H NMR (400 MHz, DMSO-d6) δ 8.20–8.08 (m, 2H), 7.74 (ddd, J = 11.6, 6.8, 1.6 Hz, 2H), 7.69–7.63 (m, 3H), 7.51–7.33 (m, 5H), 7.06 (t, J = 8.8 Hz, 2H), 5.12–5.07 (m, 2H), 4.85 (s, 1H). ^13^C NMR (151 MHz, DMSO) δ 139.74, 139.53, 133.09, 132.49, 131.88, 131.31, 131.25, 131.09, 131.04, 130.74, 130.39, 129.73, 129.65, 129.31, 128.93, 128.86, 115.70, 115.55, 76.24, 76.23, 43.56, 43.14. ^31^P NMR (243 MHz, DMSO) δ 29.97. ^19^F NMR (565 MHz, DMSO) δ -114.23.

#### 3.5.6. **3f** ((1-(4-chlorophenyl)-2-nitroethyl) diphenylphosphine oxide)

^1^H NMR (400 MHz, Chloroform-d) δ 7.99 (dd, J = 11.2, 7.6 Hz, 2H), 7.70–7.59 (m, 3H), 7.53–7.41 (m, 3H), 7.34 (dt, J = 10.8, 5.2 Hz, 2H), 7.27 (dd, J = 8.4, 1.6 Hz, 2H), 7.21 (d, J = 8.4 Hz, 2H), 5.06 (d, J = 3.2 Hz, 1H), 4.78–4.70 (m, 1H), 4.47–4.38 (m, 1H). ^13^C NMR (151 MHz, CDCl_3_) δ 134.43, 132.92, 132.34, 131.15, 131.09, 130.95, 130.89, 130.74, 130.32, 130.13, 129.48, 129.40, 129.05, 128.59, 128.51, 75.69, 75.65, 45.46, 45.04. ^31^P NMR (243 MHz, CDCl_3_) δ 29.53.

#### 3.5.7. **3g** ((1-(4-bromophenyl)-2-nitroethyl) diphenylphosphine oxide)

^1^H NMR (400 MHz, DMSO-d6) δ 8.13 (td, J = 8.0, 3.6 Hz, 2H), 7.76 (dd, J = 11.6, 7.6 Hz, 2H), 7.69–7.62 (m, 3H), 7.48–7.35 (m, 7H), 5.13–5.05 (m, 2H), 4.90–4.80 (m, 1H). ^13^C NMR (151 MHz, DMSO) δ 133.13, 132.71, 132.60, 132.19, 131.99, 131.67, 131.29, 131.10, 131.03, 130.65, 130.27, 129.75, 129.67, 128.94, 121.60, 76.00, 43.75, 43.33. ^31^P NMR (243 MHz, DMSO) δ 29.69.

#### 3.5.8. **3h** ((1-(2-chlorophenyl)-2-nitroethyl) diphenylphosphine oxide)

^1^H NMR (400 MHz, Chloroform-d) δ 8.10 (ddd, J = 11.2, 7.6, 1.6 Hz, 2H), 7.89 (d, J = 7.6 Hz, 1H), 7.74–7.64 (m, 3H), 7.49–7.39 (m, 3H), 7.34 (dt, J = 8.4, 4.0 Hz, 1H), 7.26 (dd, J = 7.6, 3.2 Hz, 2H), 7.19 (d, J = 4.0 Hz, 2H), 5.25 (dd, J = 9.2, 3.2 Hz, 1H), 5.20–5.11 (m, 1H), 4.79 (ddd, J = 13.6, 6.0, 3.2 Hz, 1H). ^13^C NMR (151 MHz, CDCl_3_) δ 134.95, 132.97, 132.26, 131.94, 131.30, 131.24, 130.95, 130.89, 130.19, 129.91, 129.71, 129.56, 129.47, 129.41, 128.42, 128.20, 128.12, 127.52, 75.39, 75.38, 41.14, 40.72. ^31^P NMR (243 MHz, CDCl_3_) δ 30.73.

#### 3.5.9. **3i** ((1-(3-chlorophenyl)-2-nitroethyl) diphenylphosphine oxide)

^1^H NMR (400 MHz, Chloroform-d) δ 8.01 (dd, J = 11.2, 7.6 Hz, 2H), 7.67 (t, J = 9.2 Hz, 3H), 7.53–7.43 (m, 3H), 7.36 (dt, J = 8.8, 4.4 Hz, 2H), 7.28–7.16 (m, 4H), 5.15–5.03 (m, 1H), 4.80–4.73 (m, 1H), 4.46–4.36 (m, 1H). ^13^C NMR (151 MHz, CDCl_3_) δ 134.61, 133.83, 132.98, 132.41, 131.19, 131.13, 130.99, 130.93, 130.01, 129.65, 129.49, 129.42, 128.57, 128.49, 127.47, 73.47, 73.45, 45.76, 45.34. ^31^P NMR (243 MHz, CDCl_3_) δ 29.82.

#### 3.5.10. **3j** (1-(3,4-dimethoxyphenyl)-2-nitroethyl) diphenylphosphine oxide

^1^H NMR (400 MHz, Chloroform-d) δ 7.99 (dd, J = 11.2, 7.6 Hz, 2H), 7.70–7.59 (m, 3H), 7.53–7.41 (m, 3H), 7.34 (dt, J = 10.8, 5.2 Hz, 2H), 7.27 (dd, J = 8.4, 1.6 Hz, 2H), 7.21 (d, J = 8.4 Hz, 2H), 5.06 (d, J = 3.2 Hz, 1H), 4.78–4.70 (m, 1H), 4.47–4.38 (m, 1H), 3.88 (s, 3H), 3.77 (s, 3H). ^13^C NMR (151 MHz, CDCl_3_) δ 134.43, 132.92, 132.34, 131.15, 131.09, 130.95, 130.89, 130.74, 130.32, 130.13, 129.47, 129.40, 129.04, 128.59, 128.51, 122.45, 122.05, 122.00, 75.67, 55.45, 55.45, 45.46, 45.04. ^31^P NMR (243 MHz, CDCl_3_) δ 29.53.

#### 3.5.11. **5a** (2-((diphenylphosphoryl)(phenyl)methyl) malononitrile)

^1^H NMR (400 MHz, Chloroform-d) δ 8.05–7.97 (m, 2H), 7.72–7.60 (m, 3H), 7.56–7.49 (m, 2H), 7.45–7.39 (m, 3H), 7.31 (td, J = 5.2, 4.8, 3.2 Hz, 5H), 4.78 (t, J = 7.6 Hz, 1H), 4.09 (s, 1H). ^13^C NMR (151 MHz, CDCl_3_) δ 134.43, 132.92, 132.34, 131.15, 131.09, 130.95, 130.89, 130.74, 130.32, 130.13, 129.47, 129.40, 129.04, 128.59, 128.51, 111.14, 111.09, 111.05, 75.67, 45.46, 45.04, 24.31. ^31^P NMR (243 MHz, CDCl_3_) δ 29.53.

#### 3.5.12. **5b** (2-((2-chlorophenyl) (diphenylphosphoryl)methyl) malononitrile)

^1^H NMR (400 MHz, DMSO-d6) δ 8.20–8.12 (m, 2H), 7.99 (d, J = 8.0 Hz, 1H), 7.76–7.64 (m, 3H), 7.50 (dd, J = 11.6, 7.6 Hz, 2H), 7.42 (t, J = 7.2 Hz, 2H), 7.36–7.26 (m, 4H), 5.66 (dd, J = 8.4, 6.8 Hz, 1H), 5.22–5.15 (m, 1H). ^13^C NMR (151 MHz, DMSO) δ 134.63, 134.19, 133.99, 133.45, 132.74, 132.28, 132.20, 131.84, 131.78, 130.97, 130.76, 130.28, 130.14, 129.68, 129.60, 129.24, 129.16, 128.81, 128.73, 113.20, 113.13, 112.01, 111.91, 42.11, 41.83, 25.05. ^31^P NMR (243 MHz, DMSO) δ 29.01.

#### 3.5.13. **5c** (2-((3-chlorophenyl) (diphenylphosphoryl)methyl) malononitrile)

^1^H NMR (400 MHz, Chloroform-d) δ 8.02 (dd, J = 11.6, 7.6 Hz, 2H), 7.75–7.63 (m, 3H), 7.59–7.44 (m, 3H), 7.36 (d, J = 7.2 Hz, 5H), 7.27 (d, J = 8.0 Hz, 1H), 4.74 (s, 1H), 4.03 (t, J = 7.6 Hz, 1H). ^13^C NMR (151 MHz, CDCl_3_) δ 135.15, 133.43, 131.32, 131.30, 131.20, 131.18, 130.53, 129.87, 129.65, 129.48, 129.41, 128.65, 128.57, 127.72, 110.94, 110.93, 47.07, 46.67, 24.62. ^31^P NMR (243 MHz, CDCl_3_) δ 28.36.

#### 3.5.14. **5d** (2-((diphenylphosphoryl)(3-nitrophenyl) methyl) malononitrile)

^1^H NMR (400 MHz, Chloroform-d) δ 8.19 (d, J = 8.8 Hz, 2H), 8.09–8.02 (m, 2H), 7.96 (d, J = 7.6 Hz, 1H), 7.71 (ddd, J = 18.8, 7.6, 2.4 Hz, 3H), 7.57 (ddd, J = 11.2, 8.0, 2.0 Hz, 3H), 7.44 (d, J = 7.2 Hz, 1H), 7.35 (td, J = 7.6, 3.2 Hz, 2H), 4.75 (s, 1H), 4.20 (t, J = 7.2 Hz, 1H). ^13^C NMR (151 MHz, CDCl_3_) δ 135.18, 133.71, 133.09, 132.91, 131.33, 131.27, 130.95, 130.89, 130.47, 129.67, 129.59, 128.91, 128.83, 125.11, 124.25, 110.68, 46.98, 46.57, 24.47. ^31^P NMR (243 MHz, CDCl_3_) δ 28.24.

#### 3.5.15. **5e** (2-((diphenylphosphoryl)(m-tolyl) methyl) malononitrile)

^1^H NMR (400 MHz, Chloroform-d) δ 8.03–7.97 (m, 2H), 7.66 (dtd, J = 14.8, 7.2, 2.4 Hz, 3H), 7.55–7.48 (m, 2H), 7.43 (dd, J = 7.6, 1.6 Hz, 1H), 7.33 (dd, J = 7.6, 3.6 Hz, 2H), 7.22–7.11 (m, 4H), 4.83 (s, 1H), 4.03 (t, J = 8.0 Hz, 1H), 2.28 (s, 3H). ^13^C NMR (151 MHz, CDCl_3_) δ 133.17, 132.69, 132.37, 131.36, 130.58, 130.30, 130.13, 129.30, 129.22, 129.12, 128.88, 128.35, 128.27, 126.77, 111.31, 111.30, 47.42, 47.00, 24.70, 21.30. ^31^P NMR (243 MHz, CDCl_3_) δ 28.64.

#### 3.5.16. **5f** (2-((diphenylphosphoryl)(3-methoxyphenyl) methyl) malononitrile)

^1^H NMR (400 MHz, Chloroform-d) δ 8.04–7.95 (m, 2H), 7.71–7.53 (m, 5H), 7.45 (d, J = 1.6 Hz, 1H), 7.33 (d, J = 3.2 Hz, 2H), 7.22 (d, J = 8.0 Hz, 1H), 7.01–6.92 (m, 2H), 6.86 (dd, J = 8.4, 2.4 Hz, 1H), 4.77 (s, 1H), 4.05 (t, J = 8.0 Hz, 1H), 3.73 (s, 3H). ^13^C NMR (151 MHz, CDCl_3_) δ 133.20, 132.46, 132.03, 131.36, 131.36, 130.31, 129.65, 129.35, 129.27, 128.85, 128.45, 128.46, 128.38, 122.08, 115.48, 114.83, 111.28, 55.29, 47.43, 47.01, 24.78. ^31^P NMR (243 MHz, CDCl_3_) δ 30.84.

#### 3.5.17. **5g** 2-((diphenylphosphoryl)(4-fluorophenyl) methyl) malononitrile

^1^H NMR (400 MHz, DMSO-d6) δ 8.17–8.09 (m, 2H), 7.69–7.62 (m, 5H), 7.53 (s, 4H), 7.44–7.32 (m, 3H), 5.46 (t, J = 7.2 Hz, 1H), 5.17 (t, J = 7.2 Hz, 1H). ^13^C NMR (151 MHz, DMSO) δ 133.11, 132.60, 132.46, 132.39, 132.32, 131.88, 131.62, 131.56, 131.37, 131.30, 130.82, 130.76, 130.16, 130.10, 129.49, 129.42, 128.87, 128.79, 116.00, 115.85, 115.09, 114.95, 113.17, 112.93, 79.57, 79.35, 79.14, 42.23, 41.80, 25.47. ^31^P NMR (243 MHz, DMSO) δ 28.58. 19F NMR (565 MHz, DMSO) δ -112.90.

#### 3.5.18. **5h** 2-((4-bromophenyl) (diphenylphosphoryl)methyl) malononitrile

^1^H NMR (400 MHz, Chloroform-d) δ 8.03–7.97 (m, 2H), 7.83–7.50 (m, 7H), 7.44 (dtd, J = 8.4, 3.9, 2.0 Hz, 3H), 7.35 (td, J = 7.6, 3.2 Hz, 1H), 7.02 (t, J = 8.6 Hz, 1H), 4.69 (t, J = 7.2 Hz, 1H), 4.05 (t, J = 7.6 Hz, 1H). ^13^C NMR (151 MHz, CDCl_3_) δ 137.03, 136.79, 133.15, 132.51, 131.97, 131.28, 130.69, 130.43, 130.25, 129.81, 129.27, 128.80, 128.45, 111.89, 111.37, 78.00, 77.95, 42.23, 41.80, 25.47. ^31^P NMR (243 MHz, CDCl_3_) δ 29.69.

#### 3.5.19. **5i** (2-((3,4-dimethoxyphenyl) (diphenylphosphoryl)methyl) malononitrile)

^1^H NMR (400 MHz, Chloroform-d) δ 8.02–7.92 (m, 2H), 7.72–7.53 (m, 5H), 7.47 (t, J = 7.6 Hz, 1H), 7.36 (td, J = 7.6, 2.8 Hz, 2H), 6.96 (d, J = 5.6 Hz, 2H), 6.82 (d, J = 8.4 Hz, 1H), 4.73 (s, 1H), 4.04–3.96 (m, 1H), 3.88 (s, 3H), 3.77 (s, 3H). ^13^C NMR (151 MHz, CDCl_3_) δ 139.74, 139.53, 133.21, 132.49, 131.88, 131.31, 131.25, 131.09, 131.04, 130.74, 130.39, 129.73, 129.65, 129.31, 128.94, 128.86, 78.00, 77.95, 57.37, 56.85, 47.42, 47.01, 24.70. ^31^P NMR (243 MHz, CDCl_3_) δ 28.47.

#### 3.5.20. **5j** (2-((diphenylphosphoryl)(naphthalen-2-yl) methyl) malononitrile)

^1^H NMR (400 MHz, Chloroform-d) δ 8.03 (dd, J = 11.6, 7.6 Hz, 2H), 7.91 (s, 1H), 7.81 (t, J = 9.2 Hz, 4H), 7.73–7.61 (m, 3H), 7.58–7.46 (m, 5H), 7.25 (td, J = 7.6, 3.2 Hz, 2H), 4.85 (t, J = 7.6 Hz, 1H), 4.23 (t, J = 8.0 Hz, 1H). ^13^C NMR (151 MHz, CDCl_3_) δ 133.23, 133.11, 132.45, 132.02, 131.35, 131.29, 130.38, 129.72, 129.49, 129.38, 129.31, 129.26, 128.79, 128.46, 128.38, 128.18, 127.72, 127.07, 126.80, 126.48, 111.36, 111.30, 111.21, 47.54, 47.12, 24.97. ^31^P NMR (243 MHz, CDCl_3_) δ 28.51.

#### 3.5.21. **5k** (2-((diphenylphosphoryl)(thiophen-2-yl) methyl) malononitrile)

^1^H NMR (400 MHz, Chloroform-d) δ 7.96 (q, J = 7.6, 6.4 Hz, 2H), 7.86–7.74 (m, 2H), 7.70 (d, J = 7.2 Hz, 1H), 7.64 (d, J = 7.6 Hz, 4H), 7.52 (s, 1H), 7.41 (d, J = 7.2 Hz, 2H), 7.25 (d, J = 3.2 Hz, 1H), 4.75 (s, 1H), 4.40 (s, 1H). ^13^C NMR (151 MHz, CDCl_3_) δ 133.37, 132.78, 132.35, 131.89, 131.64, 131.58, 131.41, 131.34, 130.80, 130.68, 129.81, 129.42, 129.34, 129.16, 128.87, 128.55, 128.47, 128.36, 128.16, 127.85, 127.45, 126.82, 111.25, 111.19, 110.96, 110.92, 42.97, 42.54, 25.62. ^31^P NMR (243 MHz, CDCl_3_) δ 28.36.

## 4. Conclusions

In conclusion, we developed a lipase-catalyzed phospha-Michael addition reaction of β-nitrostyrenes or benzylidenemalononitrile with diphenylphosphine oxide as a phosphine source under mild reaction conditions. This enzymatic method utilizes available substrates, environment-friendly solvents, and simple and practical operation (requiring only filtration and washing without column chromatography). Novozym 435 has been demonstrated to be an efficient, atom-economic, and reusable catalyst for the synthesis of various organophosphorus compounds. Furthermore, immobilization is an efficient tool for improving enzyme features in the biotechnology toolbox. To improve the feasibility and efficiency of this synthetic method, other Cal-B immobilization methods are being studied and will be reported in due course [58,59,60].

## Data Availability

Samples are not available from the corresponding authors.

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
