# Peer review of "Lipase-Catalyzed Phospha-Michael Addition Reactions under Mild Conditions"

_molecules, 2022, doi:10.3390/molecules27227798_

Round 1

Reviewer 1 Report

In this paper, wang and the cooperator developed a lipase-catalyzed phospha-Michael addition reaction of β-nitrostyrenes or benzylidenemalononitrile with diphenylphosphine oxide as a phosphine source under mild reaction conditions. Notably, this transformation has been well-established (RSC Adv., 2017, 7, 42792-42799), and the corresponding asymmetric version has been successfully achieved by several groups (Chem. Commun., 2007, 5058-5060; Org. Lett. 2013, 15, 5476–5479; Adv. Synth. Catal., 2012, 354, 3122-3126.). The present study promoted the titled Michael addition via a biocatalytic approach. The manuscript is suitable for the publication in Molecules after major revisions.

Firstly, the references mentioned above should be added in proper place.

Secondly, 13C NMR ans 31P NMr spectra are required for all compounds, 19 F NMR spectra are required for 3e and 5g.

Moreover, the proposed mechanism should be revised. The nitro group may be actived via hydrogen-bonding interaction with the catalyst.

Author Response

Thanks for your helpful suggestions, we added the details and revised the MS accordingly. The relevant files were attached.

Reviewer 2 Report

The paper has a great interest. However, discussion should be enhance, and introduction reinforce to facilitate a deeper discussion.

Abstract should mention, at least,  the best lipases for this reaction, and if they have been used in free or in immobilized form. Some numbers may be welcome (enzyme activity, productivity, purity, etc.). Any advance in mechanism?

Organ catalysts? met-al catalysts?

Introduction of enzymes require some reference (some of the recent reviews from Woodley or Sheldon groups).

Promiscuity also require introduction, there are a handful of reviews that could be used.

Lipases also need introduction, even a short one. First as interfacial enzymes (describing the mechanism of interfacial activation), second, as some of the enzymes with more promiscuous  activities.

Immobilization is used in the paper, but is introduced very slightly in results, when most of the statements could be included in introduction. Immobilization started just to solve the recovering and reuse of the enzymes, initially very expensive. The immobilized enzymes also facilitate the control of the reaction. Later, enzyme immobilization was found to be a solution for many enzyme limitations: enzyme stability could be improved (by different causes recently reviewed by Lafuente), but also can alter enzyme selectivity, specificity, activity, reduce inhibitions, enlarge the operation window and resistance to chemical, and even be coupled to enzyme purification. There are reviews in each of these topics. This can be achieved only of the enzyme immobilization is properly performed (see very recent review from Prof Woodley).

Introduce the lipases and the specific biocatalyst used, adding proper references (reviews if available)¡, justifying the selection.

“was the best 66 in generating the highest yield of the product” Redundant.

“Novozym 435 69 was inactivated by an irreversible inhibitor (PMSF) to confirm the role of the active center 70 of lipase in this phospha-Michael addition reaction” References for this? Or is it the first time that this is performed? This is very important point and should be in abstract.

They use a similar amount of protein  in all assays? How they compare Novozym 435 and CALB, this may deserve some discussion.

“An appropriate reaction medium can improve the yields of enzymatic reactions ob- 76 viously.” References?

“neither of triethylamine or lipase” word missed?

 The effect of the concentration of Novozym 435 in the final yield is curious. This should affect the reaction rate, not the yields. I guess that really they are measuring at a fixed time before reaction completion. This also explain why the use of more enzyme has not effect, they have reached the yield in the used time, however, using shorter times, perhaps they can visualize some effect.

Novozym 435 has different problems, as has been revised: enzyme leakage, support dissolution, etc. They can confirm or discard the enzyme release by comparing SDS-PAGE of the initial biocatalysts and that after 7 reuses. In any case, the decrease in activity, although significant, may be not too high and related to the loss of some biocatalysts during handling, have they weighted the final biocatalyst to ensure if they still have the same mass? The loss of mass is a problem in the lab, it should not be a problem in an industrial reactor.

In any case, an evident comment in discussion should be to search for other CALB immobilization method that could improve the enzyme stability.

Moreover, Garcia-Verdugo also has shown how the immobilization protocol can drastically affect the reaction activity in promiscuous reactions.

Author Response

(The authors gave the same response as above.)

Reviewer 3 Report

Although the authors give several references on the catalytic promiscuity of lipases, they seem to ignore recent papers on their subject, the application of lipases for the catalysis of Phospa-Michael reaction (e.g. the 2022 paper of Samsonowicz-Gorski et al. at the journal of Int. J. Mol. Sci. ). Please add more recent refs on the subject.

Author Response

(The authors gave the same response as above.)

Round 2

Reviewer 1 Report

accept

Reviewer 2 Report

Authors have properly answered my comments.